# A Network DEA Approach for Performance Evaluation of Safety Supervision and Rescue Capability in the Port Waters of Changjiang MSA

Guo-Ya Gan [1], Qin Wang [1,*] and Qian-Feng Wang [2]

1   School of Business, Nanjing Audit University, Nanjing 211815, China
2   School of Management, Shanghai University of Engineering and Science, Shanghai 201620, China
*   Correspondence: wangqin@nau.edu.cn

**Abstract:** In recent years, the global economic situation and the development of the international shipping industry have been deeply affected by COVID-19. Since 2021, China has gradually recovered its international shipping supply chain industry with the help of government policy support, and its GDP has grown by 8.1% year by year. Under this favorable macroeconomic background, the Yangtze River waterway transportation, with its good waterway conditions, has led to the continuous increase in transportation demands. However, while pursuing rapid economic growth, ensuring the navigation safety and rescue of ships in the waterway has been one of the key issues of concern for maritime divisions along the Yangtze River. Therefore, combined with the network date envelopment data (DEA) model, this study intends to construct a new set of performance evaluation models in line with their safety supervision and rescue capability based on the daily work characteristics of the Changjiang Maritime Safety Administration (MSA). The occurrence of disasters in their port areas has been taken into consideration as the key undesirable variable. This study hopes to screen out worthy pacesetter representatives, and further suggests more targeted improvement options for inefficient maritime authorities to facilitate more effective safety supervision in the future.

**Keywords:** network date envelopment data (DEA); performance evaluation; safety supervision; rescue capacity; Changjiang Maritime Safety Administration (MSA)

## 1. Introduction

In China, the Maritime Safety Administration (MSA) is a government department approved by the State Council of the People's Republic of China for the supervision of water transportation safety. The China MSA is under a vertical management system. According to the authorization of Chinese laws and regulations, the China MSA is mainly responsible for exercising national water safety supervision and prevention of ship pollution, ship and marine facilities inspection, navigation security management and administrative law enforcement, as well as performing the management functions of the Ministry of Transport of the People's Republic of China (such as safety production). Specifically, it is responsible for the formulation and organization of the implementation of national water traffic safety supervision and management, the unified management of water traffic safety and prevention of ship pollution, ship and marine facilities inspection industry management, as well as ship seaworthiness and ship technology management, the crew, pilot, magneto warp corrector fitness training, examination, licensing management and management of navigation order, navigation environment and other work matters.

Among them, the Changjiang MSA is one of the important maritime authorities directly under the Ministry of Transport of China, which is administratively subordinate to the Yangtze River Navigation Administration. This maritime authority has direct jurisdiction over a total of 2695.5 km of Yangtze River mainline waters from Hejiangmen in Yibin to Liuhekou in Jiangsu, which is known as an important transportation corridor for

the main functional areas such as the Yangtze River midstream city group, Yangtze River Delta city group and the Yangtze River economic belt in an integrated three-dimensional manner. For this reason, the Changjiang MSA has set up a total of 12 branch maritime authorities, including: Yibin, Luzhou, Chongqing, Yichang, Three Gorges, Jingzhou, Yueyang, Wuhan, Huangshi, Jiujiang, Anqing and Wuhu MSAs. These branch maritime authorities are mainly responsible for the supervision and management of water traffic safety and prevention of ship pollution under their jurisdiction, the pilotage of Yangtze River, water safety communication and maintenance of the main channel traffic on the Yangtze River and other comprehensive law enforcement work. At the same time, when major water traffic accidents occur in the waters under the jurisdiction of these maritime authorities, or when ships cause major water pollution accidents, these branch MSAs will serve as an important functional government department in the disposal, command and coordination of rescue work. They will be mainly responsible for the organization, guidance and specific implementation of the search and rescue of water traffic accidents and ship pollution accidents in the jurisdiction, as well as the comprehensive work of water traffic violations to carry out a comprehensive investigation and treatment of cases.

Due to the impact of force majeure factors as a result of the COVID-19 pandemic, the demands for waterborne freight transport on the Yangtze River route keeps plummeting. According to the survey of Yangtze River Port and Maritime Logistics Alliance, the production and operation of shipping enterprises on the Yangtze River mainline generally fell into difficulties in the first half of 2020, among which the dry bulk cargo and container traffic of the Yangtze River Group dropped by 10.9% and 16.4% year on year, respectively [1]. With the strong protection of the Chinese government for the waterway transportation of important production and work-introduced policies for living and emergency materials, since 2021, both water freight transportation and Yangtze River tourist passenger transportation market have shown a stable recovery. These effective measures have also driven the vessel traffic on the Yangtze River mainline to increase day by day. However, with the increase of vessel traffic in the main channel of Yangtze River, water traffic accidents and water pollution incidents in the jurisdiction of the Changjiang MSA have also emerged along with it. According to the report by the China Water Transport Network, in the first quarter of 2022, there were 15 water traffic accidents on the Yangtze River Main Line, including 2 general grade accidents of transport vessels, resulting in 2 deaths, 2 shipwrecks and direct economic losses of up to 3.9 million [2]. Therefore, this highlights the importance of the branch maritime authorities of Changjiang MSA in performing their safety supervision duties in the jurisdictional port area. High-quality daily ship safety supervision and timely rescue work after accidents can effectively ensure smooth traffic flow in the jurisdictional port area and minimize the loss of life and property of ships and crew. Therefore, this study aims to evaluate the performance of safety supervision and rescue capability of each branch maritime bureau of the Changjiang MSA, in order to assess the benchmark maritime authorities with satisfactory performance for other inefficient units to learn from. Furthermore, they can jointly maintain a safe navigation environment in the port area of the Yangtze River.

The remaining parts of this paper can be organized as follows: Section 2 reviews the literature on the studies about maritime safety and port management, along with network date envelopment data (DEA) applications with undesirable outputs. Then, a network DEA model taking into account the undesired outputs is introduced in Section 3, which briefly describes the process of using the DEA model. Then, a real-world case example of Changjiang MSA in 2022 is presented in Section 4. Finally, Section 5 concludes the evaluation results and discusses the managerial implications.

## 2. Literature Review

### 2.1. Maritime Safety and Port Management

Research on maritime safety supervision and port management has always been one of the more important issues concerning relevant scholars in recent years. These scholars

have conducted studies from different perspectives with the aim of helping port managers to maintain the smooth flow of water traffic in the port area and to ensure the safety of both ships and crew [3–8].

In order to reduce the navigational collision risk of ships in port areas, Ozturk et al. [9] proposed three new parameters (distance, area and ship speed) to improve the current navigation collision prevention risk model applicable to the port basins. At the same time, they also constructed a new risk assessment method based on fuzzy reasoning and machine learning. Considering the impact of climate change on ship operations in ports, Camus et al. [10] developed a hybrid statistical–dynamic framework model based on weather generators and metamodels. Their new model can help predict the future hourly sea conditions in the port area once the ship enters the port, and this information will be effective in helping ship operators to navigate safely in the port area waters. Makris et al. [11] used three robust numerical models to simulate the motion of wave fields in the harbor area of bays and large ports, with the aim of developing applications to assist ships in berthing at ports. They hoped to help the ships berth safely in different weather conditions. Mainly from the perspective of relevant regulations on air emissions in port waters, Schinas [12] explored the impact of international laws on port managers' decision-making and operations. His study also suggested some effective solutions to the problems identified in the daily management of ports. Jeong et al. [13] conducted a safety assessment study on the risk of potential gas leakage from the filling system of LNG vessels, using the world's first 50,000-ton LNG vessel as the research subject of their study. They also pointed out the shortcomings and deficiencies of existing free trade areas (FTAs) in terms of existing technical and regulatory guidance, and proposed targeted management response strategies in anticipation of providing practical reference for port authorities and flag state managers, whereas Aneziris et al. [14] developed a unique training program for the operation procedures and management norms of LNG pressurized tanks in the port area, with the purpose of helping relevant managers (such as port operators, maritime instructors, etc.) in the port area to master the safety knowledge of LNG fuel chain. Based on the traditional literature review method, Chuah et al. [15] selected the data from the Maritime Safety Administration inspection from 2016 to 2021, and combined them with qualitative and quantitative analyses, in order to explore the relationship between various risk factors and safety accidents in the port area. Their research results provide important enlightenment on how to effectively select the ships to be inspected, and further strengthen the risk prevention and control of port safety and ship safety. Due to the differences between the world's flag states in all Port State Control (PSC), Kara [16] pointed out that it results in varying levels of maritime security for the world's trade fleet. They used the TOPSIS methodology to measure the overall performance of the trade fleet under different flag states and found that nearly half of the flag states had overall performance levels below the international standards. International long-distance multimodal transport usually includes multiple modes of transportation, but the most critical transportation service to achieve global trade flows is the container drayage operation within the port area. Chen et al. [17] provided a comprehensive review of the extant research literature related to container port drayage apparatus from different algorithm design and technology perspectives, and outlined some responses to the safety management for stakeholders in the port area.

### 2.2. Network DEA with Undesirable Outputs

DEA is an evaluation method used in the field of operations research and economics to explore production frontiers, which was first proposed by Charnes et al. [18] in 1978, and this method is generally used to measure the productivity of some decision sectors. Then, Banker et al. [19] extended their DEA model from constant return to scale (CRS) to cover variable returns to scale (VRS). Since then, DEA models have been extended to other practical domains in the form of super-efficiency models [20–22], cross-efficiency models [23,24], SBM models and super-SBM models [25–28], DEA models with uncertain inputs or outputs [29]. In recent years, as the complexity of production management

structure has developed, the models for assessing operational production performance have also developed toward to the network structure. Many researchers begin to expand the traditional DEA models into the network DEA models, and gradually consider some undesired outputs as the special variables in their studies [30–34].

The traditional DEA model treats decision-making units (DMUs) as black boxes, which may ignore the interactions between different processes. Sarkhosh-Sara et al. [35] found that using the existing network DEA model often leads to model error reporting when dealing with zero data or undesirable outputs. For this reason, they proposed a new network DEA model (NDEA), and used it to assess the sustainability indices of countries with different development status to determine the strengths and weaknesses of each country. Then, they provided some response options for inefficient countries when planning sustainable production and resource reallocation. In contrast, Pishgar-Komleh et al. [36] focused their research on the production aspect of agriculture. They integrated the life cycle assessment (LCA) methods and DEA to assess the production performance status of winter wheat cropping systems in Poland. According to comparison results of six models, they found that a slack-based DEA model with considering undesirable outputs can better reflected the producing performance of winter wheat cropping. Further, the assessment results could not only maximize the production efficiency of winter wheat but also minimize the generation of undesirable products. Mozaffari et al. [37] pointed out that the traditional network DEA model can only evaluate performance with explicit input and output variables, for this reason, they innovatively combined triangular fuzzy numbers for the traditional model improvement and proposed a new two-stage DEA model to deal with the multi-objective linear programming. In addition, their new model can be applied not only in the presence of imprecise data but also carry out effective performance evaluation for DMUs with undesirable output variables (such as GHG emissions, hazardous wastewater, etc.) Moreover, the business of many enterprises is completed by the linkage of multiple departments, and the intermediate measures can often be transformed into the final products in the subsequent stage. In this complex production network structure, many uncertainties should be considered as the important topics when conducting assessments of the management performance of business operations. To this end, Omrani et al. [38] extended the two-stage network DEA model to handle scenarios with negative data and undesirable outputs through the non-linear programming techniques (e.g., dichotomous search). Finally, they used a real case, including 22 insurance companies to verify the validity and realistic value of the new model. In addition, the green productivity of enterprises has been one of the most important indicators for the development of green economy by the government. Qu et al. [39] used sulfur dioxide, nitrogen oxides and soot as newly incorporated undesirable outputs, combined with the entropy weighting method to measure the environmental pollution green development indices of industries and cities in Zhejiang Province. Finally, they applied these indices to an output-oriented DEA model to measure the global Malmquist-Luenberger productivity index (GMLPI).

*2.3. Discussion*

Through the above literature review, it can be seen that maintaining the safety supervision and safeguarding the environment from pollution in the port waters has always been an important regulatory responsibility of government MSA. Additionally, this research issue has also attracted a lot of attention from scholars in the related fields. At the same time, although the network DEA models have been continuously expanded and improved, and many scholars have applied these new expanded models with considering the undesirable outputs to different industries, the real application of port safety regulation and rescue performance evaluation is relatively rare. Therefore, this study aims to evaluate the performance of maritime safety supervision and relief capabilities in each branch maritime bureau of the Jangjiang MSA in China by combining the network DEA model, taking into account the undesirable outputs.

## 3. Methodology

The network DEA model used in this study is described via the sets of indices, variables and parameters presented in Table 1.

**Table 1.** Nomenclature.

| | | |
|---|---|---|
| Index | $j = 1, \ldots, n$ | DMU (port crane equipment) |
| | $i = 1, \ldots, m$ | Inputs |
| | $g = 1, \ldots, h$ | Intermediate products |
| | $r = 1, \ldots, s$ | Good outputs |
| | $t = 1, \ldots, q$ | Undesirable outputs |
| Variables | $\lambda_j$ | Weights of peers (DMU-$j$) in stage 1 |
| | $w_j$ | Weights of peers (DMU-$j$) in stage 2 |
| | $x_{ij}$ | Input $i$ of DMU-$j$ |
| | $z_{gj}$ | Intermediate products $g$ of DMU-$j$ |
| | $y_{rj}$ | Output $r$ of DMU-$j$ |
| | $y_{tj}^u$ | Undesirable output $t$ of DMU-$j$ |
| | $v_i$ | Non-negative intensity vectors of input $i$ |
| | $w_g$ | Non-negative intensity vectors of intermediate products $g$ |
| | $u_r$ | Non-negative intensity vectors of good output $r$ |
| | $d_t$ | Non-negative intensity vectors of undesirable output $t$ |
| Parameters | $\theta_j^*$ | Overall efficiency of DMU-$j$ |
| | $\theta_{1j}^*$ | Efficiency of DMU-$j$ in stage 1 |
| | $\theta_{2j}^*$ | Efficiency of DMU-$j$ in stage 2 |
| | $E_{\max}$ | Maximum value of $\theta_j^*$ |
| | $TSCI_j$ | Total safety capacity index of DMU-$j$ |

A basic straight two-stage network structure with intermediate products as shown in Figure 1, should be considered for each of a set of n DMUs. Compared with the traditional DEA approaches, the intermediate productions during the black box are considered. Suppose that each DMU has a set of $m$ input variables $x_{ij}$ ($i =1, 2, \ldots, m$) and $s$ output variables $y_{rj}$ ($r = 1, 2, \ldots, s$), respectively. In addition, a set of $h$ intermediate variables $z_{gj}$ ($g =1, 2, \ldots, h$) produced in the first stage (sub-process 1) will flow to the second stage (sub-process 2). Then, the production possible set (PPS) of each DMU can be described as follows:

$$T_p = \left\{ (x_i^p, y_r^p, z_g^p) \middle| \begin{array}{c} \exists \lambda_j^p \in \Lambda_p \forall j; \sum_{j=1}^n \lambda_j^p x_{ij}^p \leq x_{ik}^p \ \forall i \in I(p); \ \sum_{j=1}^n \lambda_j^p y_{rj}^p \geq y_{rk}^p \ \forall r \in O(p) \\ z_{gk}^p \geq \sum_{j=1}^n \lambda_j^p z_{gj}^p \ \forall g \in G^{in}(p) \ ; z_{gk}^p \leq \sum_{j=1}^n \lambda_j^p z_{gj}^p \ \forall g \in G^{out}(p) \end{array} \right\} \tag{1}$$

where $\lambda_j^p (j = 1, \ldots, n)$ are nonnegative intensity vectors, and as in conventional network DEA, the set $\Lambda_p$ denote the returns to scale (RTS) assumption for process $p$.

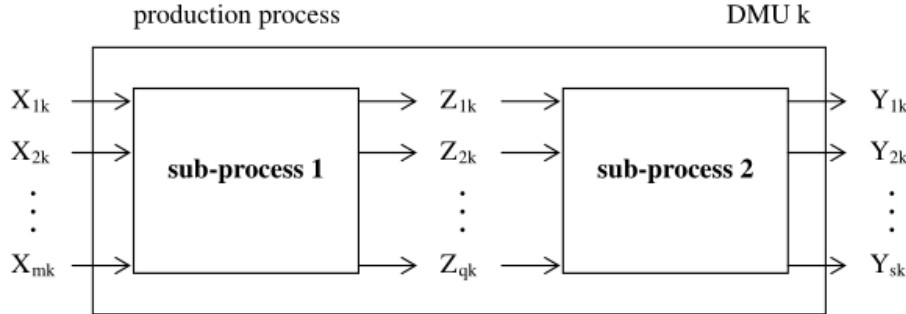

**Figure 1.** A two-stage network structure with intermediate products.

This network structure is a simple two-stage structure, which is usually named as the basic network system in DEA applications. To measure the efficiency of this two-stage network structure under constant returns to scale, Kao [40] described the efficiency formulation in its envelopment form:

$$
\begin{aligned}
Max \quad & \theta \\
s.t. \quad & \sum_{j=1}^{n} \lambda_j x_{ij} \leq \theta x_{ik}, \ i = 1, \ldots, m \ ; \\
& \sum_{j=1}^{n} \mu_j y_{rj} \geq y_{rk}, \ r = 1, \ldots, s \ ; \\
& \sum_{j=1}^{n} \lambda_j z_{gj} \geq \sum_{j=1}^{n} \mu_j z_{gj}, \ g = 1, \ldots, h \ ; \\
& \lambda_j, \mu_j \geq \varepsilon, j = 1, \ldots, n.
\end{aligned}
\tag{2}
$$

where $\lambda_j$ and $\mu_j$ denote the intensity vectors in the two stages, and $\varepsilon$ is a small non-Archimedean number to limit the intensity multipliers $\lambda_j$ and $\mu_j$, which is in order to avoid the unfavorable factors being ignored [41].

Equation (2) is in the envelopment form, which has a dual. The input-oriented basic network DEA model, proposed by Kao and Hwang [42] for evaluating DMU-k in its multiplier form, can be expressed as:

$$
\begin{aligned}
Max \quad & \frac{\sum_{r=1}^{s} u_r y_{rk}}{\sum_{i=1}^{m} v_i x_{ik}} \\
s.t. \quad & \frac{\sum_{g=1}^{h} w_g z_{gj}}{\sum_{i=1}^{m} v_i x_{ij}} \leq 1, \ j = 1, \ldots, n \ ; \\
& \frac{\sum_{r=1}^{s} u_r y_{rj}}{\sum_{g=1}^{h} w_g z_{gj}} \leq 1, \ j = 1, \ldots, n \ ; \\
& v_i, \ w_g, u_r \geq \varepsilon, i = 1, \ldots, m \ ; \\
& g = 1, \ldots, h \ , \ r = 1, \ldots, s.
\end{aligned}
\tag{3}
$$

where $v_i(i = 1, \ldots, m)$, $w_g(g = 1, \ldots, h)$ and $u_r(r = 1, \ldots, s)$ are the corresponding positive weights of the variables for each stage. Equation (3) is a non-linear programming model, which can be directly transformed into a linear programming using the Charnes–Cooper transformation [43].

$$
\begin{aligned}
Max \quad & \sum_{r=1}^{s} u_r y_{rk} \\
s.t. \quad & \sum_{i=1}^{m} v_i x_{ik} = 1 \ ; \\
& \sum_{g=1}^{h} w_g z_{gj} - \sum_{i=1}^{m} v_i x_{ij} \leq 0, \ j = 1, \ldots, n \ ; \\
& \sum_{r=1}^{s} u_r y_{rj} - \sum_{g=1}^{h} w_g z_{gj} \leq 0, \ j = 1, \ldots, n \ ; \\
& v_i, \ w_g, u_r \geq \varepsilon, i = 1, \ldots, m \ ; \\
& g = 1, \ldots, h \ , \ r = 1, \ldots, s.
\end{aligned}
\tag{4}
$$

Based upon the input-oriented basic DEA model of Charnes et al. [18], we define two ratio formulations as the efficiency of these two stages. Usually, the overall efficiency of the

network structure will be defined as the product of these two ratio efficiencies. Thus, the efficiency of DMU-k can be calculated as:

$$
\theta_k^* = \frac{\sum\limits_{r=1}^{s} u_r^* y_{rk}}{\sum\limits_{i=1}^{m} v_i^* x_{ik}} = \sum\limits_{r=1}^{s} u_r^* y_{rk}
$$

$$
\theta_{1k}^* = \frac{\sum\limits_{g=1}^{h} w_g^* z_{gk}}{\sum\limits_{i=1}^{m} v_i^* x_{ik}} = \sum\limits_{g=1}^{h} w_g^* z_{gk} \tag{5}
$$

$$
\theta_{2k}^* = \frac{\sum\limits_{r=1}^{s} u_r^* y_{rk}}{\sum\limits_{g=1}^{h} w_g^* z_{gk}}
$$

where $v_i^*(i = 1,\ldots,m)$, $w_g^*(g = 1,\ldots,h)$ and $u_r^*(r = 1,\ldots,s)$ are the optimal solution under Equation (4). $\theta_k^*$ is calculated as the optimal efficiency of the overall network system, $\theta_{1k}^*$ and $\theta_{2k}^*$ are calculated as the optimal efficiencies of the two process stages, respectively.

However, in this study, the undesirable output variables are also taken into account. Based upon Equation (1), the PPS of each DMU is redefined as follows:

$$
T_p = \left\{ (x_i^p, y_r^p, y_r^{up} z_g^p) \left| \begin{array}{l} \exists \lambda_j^p \in \Lambda_p \forall j; \ \sum\limits_{j=1}^{n} \lambda_j^p x_{ij}^p \leq x_{ik}^p \ \forall i \in I(p); \\[2ex] \sum\limits_{j=1}^{n} \lambda_j^p y_{rj}^p \geq y_{rk}^p \ \forall r \in O(p); \ \sum\limits_{j=1}^{n} \lambda_j^p y_{hj}^p \geq y_{hk}^u{}^p \ \forall h \in O^u(p) \\[2ex] z_{gk}^p \geq \sum\limits_{j=1}^{n} \lambda_j^p z_{gj}^p \ \forall g \in G^{in}(p) \ ; z_{gk}^p \leq \sum\limits_{j=1}^{n} \lambda_j^p z_{gj}^p \ \forall g \in G^{out}(p) \end{array} \right. \right\} \tag{6}
$$

Similarly, $\lambda_j^p (j = 1,\ldots,n)$ are nonnegative intensity vectors, and the set $\Lambda_p$ also denote RTS assumption for the process $p$. $O^u$ denotes the possible set of undesirable outputs. Then, for formulating the efficiency of DMU-k in this study, Equation (4) is also modified in this study, which can be described as:

$$
\begin{aligned}
Max \quad & \sum\limits_{r=1}^{s} u_r y_{rk} - \sum\limits_{t=1}^{q} d_t y_{tk}^u \\
s.t. \quad & \sum\limits_{i=1}^{m} v_i x_{ik} = 1 ; \\
& \sum\limits_{g=1}^{h} w_g z_{gj} - \sum\limits_{i=1}^{m} v_i x_{ij} \leq 0, j = 1,\ldots,n ; \\
& \sum\limits_{r=1}^{s} u_r y_{rj} - \sum\limits_{g=1}^{h} w_g z_{gj} \leq 0, j = 1,\ldots,n ; \\
& \sum\limits_{g=1}^{h} w_g z_{gj} - \sum\limits_{r=1}^{s} d_t y_{tj}^u \leq 0, j = 1,\ldots,n ; \\
& v_i,\ w_g, u_r, d_t \geq \varepsilon \quad i = 1,\ldots,m \ \ g = 1,\ldots,h ; \\
& r = 1,\ldots,s \quad t = 1,\ldots,q.
\end{aligned} \tag{7}
$$

where $\varepsilon$ is a small non-Archimedean number to impose upon the intensity multipliers $v_i$, $w_g$, $u_r$ and $d_t$. Suppose $v_i^*(i = 1,\ldots,m)$, $w_g^*(g = 1,\ldots,h)$, $u_r^*(r = 1,\ldots,s)$ and

$d_t^*(t = 1, \ldots, q)$ are the optimal solution of Equation (7), then, the overall efficiency and the decomposition efficiency of each stage for DMU-k can be calculated as follows:

$$\theta_k^* = \frac{\sum\limits_{r=1}^{s} u_r^* y_{rk} - \sum\limits_{t=1}^{q} d_t^* y_{tk}^u}{\sum\limits_{i=1}^{m} v_i^* x_{ik}} = \sum\limits_{r=1}^{s} u_r^* y_{rk} + \sum\limits_{t=1}^{q} d_t^* y_{tk}^u$$

$$\theta_{1k}^* = \frac{\sum\limits_{g=1}^{h} w_g^* z_{gk}}{\sum\limits_{i=1}^{m} v_i^* x_{ik}} = \sum\limits_{g=1}^{h} w_g^* z_{gk} \quad (8)$$

$$\theta_{2k}^* = \frac{\sum\limits_{r=1}^{s} u_r^* y_{rk} + \sum\limits_{t=1}^{q} d_t^* y_{tk}^u}{\sum\limits_{g=1}^{h} w_g^* z_{gk}}$$

In fact, when the negative results occur, we can also use the envelopment form to calculate the efficiencies of these two stages. Furthermore, in order to ensure the convenience of using the proposed model, this study draws an unambiguous flowchart in Figure 2. The flowchart for DEA application usage is as follows:

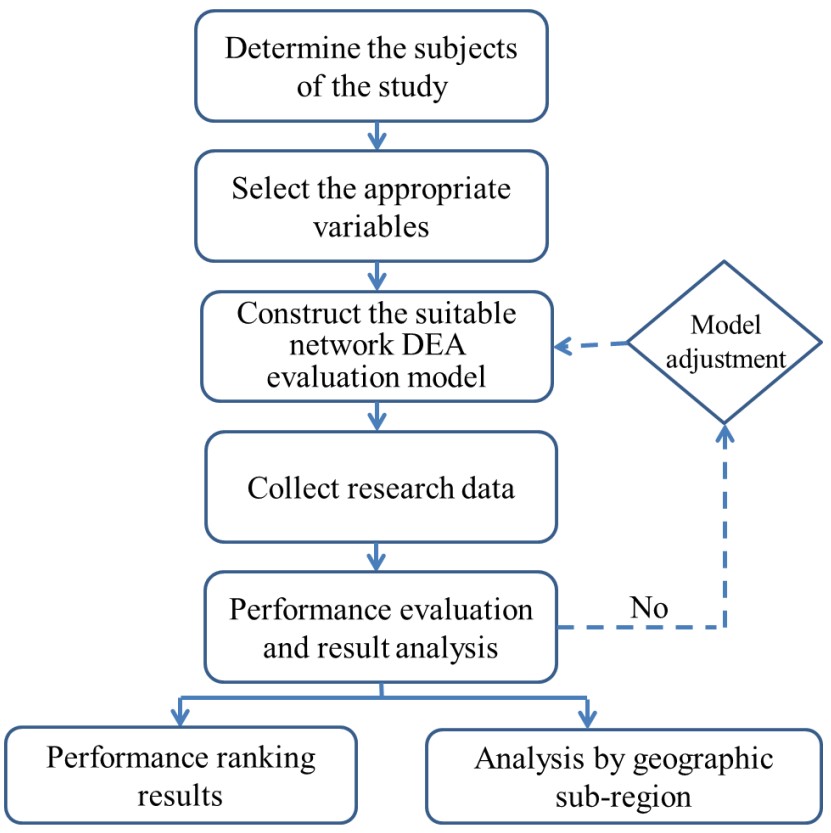

**Figure 2.** Flowchart for applying DEA model usage.

Step 1: According to the research issue, this study should determine the research subjects at the first step.

Step 2: The appropriate variables should be carefully selected based on the characteristics of the research topic and literature review. Of course, it is also important to consider some innovative variables in a specific topic.

Step 3: For the performance evaluation, this study should construct a suitable DEA model for the network structure.

Step 4: Based upon the previous steps, this study should interview some official website and some relevant managers to collect the practical research data of each variable.

Step 5: Then, this study can use the Equations (7) and (8) in this study to access the performance evaluation processing on the research data. If the infeasibility problem occurs, we should go back to Step 3 and make some suitable adjustments for the proposed DEA model, then, repeat Step 5.

Step 6: Each evaluated DMU can obtain both overall efficiency and the decomposed efficiency in this step. Then, the performance ranking results is produced and the efficient benchmarking DMUs are also selected.

Step 7: At the same time, in order to investigate the performance level of safety supervision and rescue capacity within different waters of the Yangtze River, this study will further analyze the performance level of the sub-region of Yangtze River based on the geographical location. Then, synthesizing the above findings, this study will propose some corresponding improvement suggestions for the relevant managers.

## 4. Empirical Research

### 4.1. Data Collection and Description of Variables

This study focuses on the performance assessment of the safety supervision and rescue capacity in the port waters of Changjiang MSA in China. For this reason, the data required for this study were obtained mainly through the official website of the Changjiang MSA [44] and interviews with the managers of the functional departments. The data collected for this study are presented from January to October in 2022. Due to the incompleteness of official data, two branch maritime authorities were removed from this study (Yibin and Luzhou). Therefore, in total, 10 branch maritime authorities of the Changjiang MSA were evaluated in this study, including: Chongqing, Yichang, Three Gorges, Jingzhou, Yueyang, Wuhan, Huangshi, Jiujiang, Anqing and Wuhu, respectively.

For the DEA applications, combining the characteristics of the assessed object and screening the appropriate research variables are often very important issues. Through the expert interviews, five main variables were selected in this study, including two input variables (maritime dispatch agency and jurisdictional waters), one intermediate product (vessel flow), one good output (number of rescuers) and one undesirable output (number of disaster accidents), respectively. The specific descriptions and research data are summarized in Tables 2 and 3.

**Table 2.** Descriptions for selected variables in this study.

| Name | Descriptions |
|---|---|
| Maritime dispatch agency | • This input variable is the number of dispatched law enforcement agencies under each branch MSA, whose daily work is to supervise the safety of ships navigating in the port water. |
| Jurisdictional waters | • This input variable refers to the length of port area waters under the jurisdiction of each branch maritime authority. The longer the management length is, the larger the vessel flow will be. As a result, the possibility of disaster in the port waters would be higher. |
| Vessel flow | • The intermediate product is closely related to the performance status in two different phases. In the first stage, if the value of this variable is larger, it indicates that the safety factor is high in the waters under the supervision of each branch maritime authority. The smooth navigation environment can also promote the increase in vessel traffic. On the other hand, in the second stage, the increase in vessel flow may, in turn, lead to a high number of catastrophic accidents. Therefore, this variable is screened as the intermediate product in this study. |

**Table 2.** *Cont.*

| Name | Descriptions |
|---|---|
| No. of rescuers | • When a ship traffic accident occurs in the poet waters, the dispatch agency of each branch MSA will immediately carry out the rescue work for both the ship and crew. Thus, the number of successful rescuers can be regarded as the satisfactory output of each evaluated branch maritime authority of Changjiang MSA. |
| No. of disaster accidents | • The occurrence of ship catastrophes in the port waters is an important indicator for assessing the regulatory performance of each branch MSA. We collected the number of maritime accidents that occurred in the port waters in the past two years. In principle, the maritime authorities do not expect disaster accidents or water pollution incidents to occur in the port waters; for this reason, this variable is selected as an undesirable output in this study. |

**Table 3.** Research data.

| DMUs | Input | | Intermediate Product | Output | Undesirable Output |
|---|---|---|---|---|---|
| | Dispatch Agency | Jurisdictional Waters (km) | Vessel Flow | No. of Rescuers | No. of Disaster Accidents |
| Chongqing | 13 | 681 | 1205 | 20 | 10 |
| Yichang | 5 | 127 | 1944 | 22 | 8 |
| Sanxia | 2 | 59 | 2069 | 3 | 1 |
| Jingzhou | 4 | 172 | 1798 | 10 | 6 |
| Yueyang | 5 | 162 | 3184 | 7 | 4 |
| Wuhan | 7 | 225 | 2771 | 10 | 3 |
| Huangshi | 5 | 140 | 4709 | 12 | 4 |
| Jiujiang | 5 | 128 | 6249 | 9 | 5 |
| Anqing | 6 | 243 | 9597 | 25 | 8 |
| Wuhu | 6 | 175 | 14,409 | 45 | 13 |

*4.2. Evaluation Results*

The total evaluation results of this study are summarized in Table 4. Columns 2 and 3 present the decomposed efficiency of two stages in the network structure, in terms of the safety supervision and rescue capacity, respectively. Additionally, column 4 shows the overall efficiency of each evaluated branch maritime authority of Changjiang MSA, which is the product of the value of columns 2 and 3. Due to the characteristics of network DEA models, there is often no evaluated DMU with an efficiency value of 1 in the final overall performance. Thus, to better understand the assessment results, this study defines a new total safety capacity index ($TSCI_j$), which is obtained by first calculating the overall efficiency $\theta_j^*(j = 1, \ldots, n)$ of DMU-j; then, the maximum value of $E_{\max} = \max\limits_{j=1}^{n}\left\{\theta_j^*\right\}$ can be found. Finally, the new index $TSCI_j$ can be calculated as $TSCI_j = \frac{\theta_j^*}{E_{\max}}(j = 1, \ldots, n)$. Therefore, the last two columns provide the final results of $TSCI_j$ and its corresponding rank, respectively.

**Table 4.** Research data and evaluation results in this study.

| DMUs | Efficiency of Safety Supervision of Stage 1 $\theta_{1j}^*$ | Efficiency of Rescue Capability of Stage 2 $\theta_{2j}^*$ | Overall Efficiency $\theta_j^*$ | Total Safety Capacity Inde TSCI$_j$ | Rank |
|---|---|---|---|---|---|
| Chongqing | 0.1246(10) | 0.0386 (9) | 0.0048 | 0.0143 | 10 |
| Yichang | 0.4646 (2) | 0.1862 (3) | 0.0865 | 0.2566 | 4 |
| Sanxia | 1.0000 (1) | 0.1116 (6) | 0.1116 | 0.3311 | 2 |
| Jingzhou | 0.4051 (5) | 0.0628 (7) | 0.0254 | 0.0755 | 7 |
| Yueyang | 0.3642 (6) | 0.0352(10) | 0.0128 | 0.0380 | 9 |
| Wuhan | 0.2622 (9) | 0.1650 (5) | 0.0433 | 0.1283 | 6 |
| Huangshi | 0.4214 (4) | 0.1858 (4) | 0.0783 | 0.2323 | 5 |
| Jiujiang | 0.4609 (3) | 0.0515 (8) | 0.0238 | 0.0705 | 8 |
| Anqing | 0.2700 (8) | 0.3782 (2) | 0.1021 | 0.3029 | 3 |
| Wuhu | 0.3371 (7) | 1.0000 (1) | 0.3371 | 1.0000 | 1 |
| Ave * | **0.4110** | **0.2215** | **0.0826** | **0.2449** | |

Ave * = the average value of each column.

According to the results of comprehensive index $TSCI_j$, the top 3 branch maritime authorities are Wuhu, Sanxia and Anqing MSAs. Among them, the overall performance of Wuhu MSA is much higher than other evaluated units, and it can be regarded as a learning benchmark for its peers. On the other hand, combined with the average value of evaluation performance of $TSCI_j$, it is not difficult to find that only four evaluated DMUs obtain the indices that exceeded the average level. Even, this average performance is an unsatisfactory inefficient performance result.

In addition, analyzing the performance of each branch MSA from the perspective of decomposition efficiency, it can be found that: from the perspective of safety supervision, the top three maritime authorities are Sanxia, Yichang and Jiujiang, which can better ensure the relative safety of navigation environment in the waters under management. Regarding rescue capacity, the top three maritime authorities are Wuhu, Anqing and Yichang, respectively; it is notable that only two MSA branches have exceeded the average level in this field.

*4.3. Analysis by Geographic Sub-Region*

According to the final results of the comprehensive index $TSCI_j$ in Table 4, we draw a trend graph of TSCI in Figure 3 based upon the geographical location of the Yangtze River. In China, the upper and middle reaches of the Yangtze River are divided by the city of Yichang in Hubei Province, whereas the middle and lower reaches of the Yangtze River are divided by the city of Jiujiang in Jiangxi Province.

Therefore, this study also follows this general geographical classification, and divides the evaluated branch MSAs into three categories. Then, by calculating the average performance of each region, it is found that the downstream region will obtain relatively higher overall performance than the others, whereas the midstream has shown the lowest performance level.

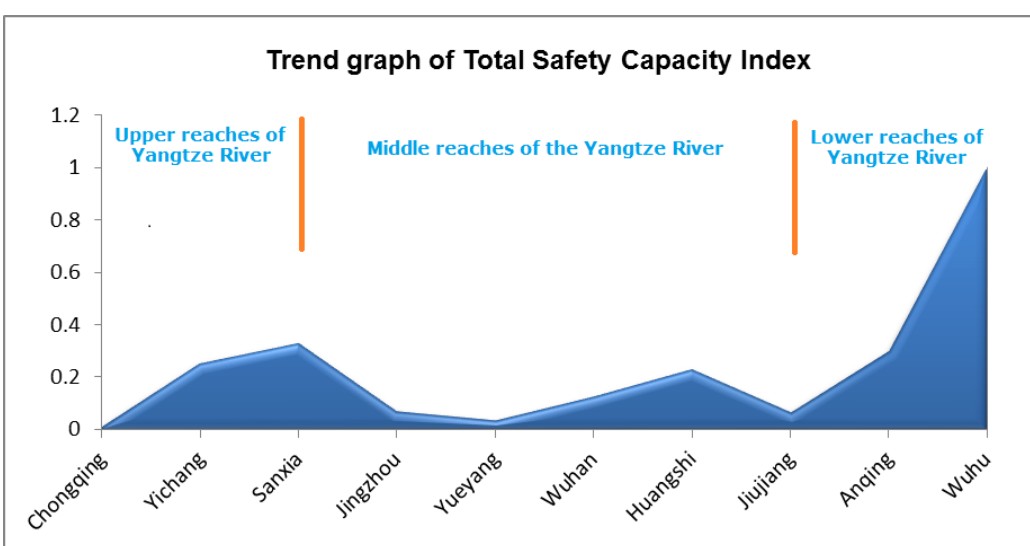

**Figure 3.** Trend graph of TSCI (by geographical location).

## 5. Conclusions and Future Directions

### 5.1. Conclusions

In China, MSA is an important part of its national comprehensive transportation and the most important administrative enforcement force on its water. All of the Chinese MSAs are charged with the important responsibility of ensuring water traffic safety, protecting the cleanliness of the waters and the overall rights of the crew, and safeguarding national maritime sovereignty. At the same time, the Yangtze River plays an important role in the economic development of this country. The Yangtze River, as the first major river running east–west in China, has been the main channel for trade and commerce since ancient times and is known as the "Golden waterway". Meanwhile, it is also one of the busiest inland waterways in the world.

Although the development of the Yangtze River shipping industry has slowed down in recent years, due to the impact of the COVID-19 epidemic, the flow of ships sailing in the main channel has recently been gradually restored with the help of government policies. To this end, ensuring the maintenance of a smooth navigation environment in the busy port waters and conscientiously performing the work functions of safety supervision and rescue has become one of the important issues of concern to the Changjiang MSA.

Based on the above discussion, this study focuses on evaluating the performance of the branch maritime authorities of Changjiang MSA at the level of safety supervision and rescue capacity. Three main issues are identified through this study, which are described as follows: (i) according to the new index of $TSCI_j$, Wuhu MSA is relatively more efficient in terms of its comprehensive performance, and it has achieved a relatively great advantage in rescue ability level, which is worthy of serving as a learning benchmark for other evaluated peers; (ii) through the analysis of efficiency decomposition, the common problem of many inefficient evaluated units is the low performance at the rescue capacity level; (iii) based upon the geographical location of the Yangtze River, the overall safety supervision and rescue capacity performance of the lower reaches of the Yangtze River is better, whereas those of the middle reaches of the Yangtze River are relatively poor.

### 5.2. Future Directions

Future research can be expanded on the basis of this study to evaluate the performance of the safety supervision and rescue capacity of Chinese coastal maritime authorities. Additionally, other mathematical research methods can be combined and some new variables with distinct characteristics can be elicited to expand the scope of this study.

**Author Contributions:** Conceptualization, G.-Y.G.; methodology, G.-Y.G.; software, Q.-F.W.; validation, Q.W.; formal analysis, G.-Y.G.; investigation, Q.W.; resources, Q.W.; data curation, Q.-F.W. and Q.W.; writing—original draft preparation, G.-Y.G.; writing—review and editing, G.-Y.G. and Q.-F.W.; visualization, G.-Y.G.; supervision, G.-Y.G.; project administration, Q.W.; funding acquisition, G.-Y.G. and Q.W. All authors have read and agreed to the published version of the manuscript.

**Funding:** This research is partially supported by National Natural Science Foundation of China (No. 72171122).

**Institutional Review Board Statement:** Not applicable.

**Informed Consent Statement:** Not applicable.

**Data Availability Statement:** Not applicable.

**Acknowledgments:** Authors are grateful to the comments and suggestions provided by anonymous reviewers.

**Conflicts of Interest:** The authors declare no conflict of interest.

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
