# Peer review of "A Network DEA Approach for Performance Evaluation of Safety Supervision and Rescue Capability in the Port Waters of Changjiang MSA"

_jmse, doi:10.3390/jmse10122002_

Round 1

Reviewer 1 Report

It is a well written paper. I have only minor comments:

1. The authors should consider analyzing further input and output orientation of the results. Moreover, I would consider the approach of Wen et al. (2014) https://doi.org/10.1155/2014/307108 where input/output are practically uncertain.

2. In the literature review, the inclusion of MCDM and marine safety ( https://scholar.google.com/citations?view_op=view_citation&hl=en&user=WB_VlEUAAAAJ&citation_for_view=WB_VlEUAAAAJ:qjMakFHDy7sC ) is missing

3. I think the authors should consider addressing the issue of air pollution and movements in the port zone - this work could be of assistance: https://doi.org/10.1007/978-3-030-39990-0_10 

Reviewer 2 Report

Congratulations by your research work this is a very interesting manuscript, almost ready to be published, only after minor corrections. Authors experience and technical skills are on display. The manuscript looks well balanced throughout the whole documents of 16 pages, I enjoyed the reading. Please consider the following comments:

Abstract: seems good, brief, clear and concise including every part of the research.

Introduction and literature review: I enjoyed the Introduction, looks appealing, however I was surprised any references were cited here. O the opposite the literature review looks sharp.

Methods: this section is interesting, technical skills are on display; some details regarding the appeal on the section could improve it, doing it sharper, as it is, it looks a bit unfinished (numbers of equations, spaces between them, sizes captions, etc.).

Results-Analysis. No doubt, authors technical skills are on full display. The quality and content are part of a deep research. A small detail is on Fig. 3 regarding the title of it compared to the name inside the figure itself.

Conclusions. Sharp and concise, including future venues of research, good job.

References. The quantity and relevancy of them seems correct.

Again, congratulations for your effort, hopefully these comments will help to improve your manuscript.

Reviewer 3 Report

I think the authors should rewrite some formulas because the text has slid below (for example formula (2),(3),(4),(5))

Also the Flowchart in Figure 2 is not easily understood (I think it needs another decisional block)
